# Dietary Betaine and Fatty Acids Change Circulating Single-Carbon Metabolites and Fatty Acids in the Dog

**DOI:** 10.3390/ani12060768

**Published:** 2022-03-18

**Authors:** Dennis E. Jewell, Matthew I. Jackson

**Affiliations:** 1Department of Grain Science and Industry, Kansas State University, 1301 Mid Campus Drive, Manhattan, KS 66506, USA; 2Hill’s Pet Nutrition, P.O. Box 1658, Topeka, KS 66601, USA; matthew_jackson@hillspet.com

**Keywords:** dogs, alpha-linolenic acid, EPA, DHA, betaine

## Abstract

**Simple Summary:**

The dog successfully consumes a wide range of dietary polyunsaturated fatty acids (PUFAs), while specific changes in fatty acid intakes have relatively unknown subsequent effects on circulating concentrations of those fatty acids and their metabolites. Similarly, single-carbon metabolism in the dog is known to be an essential part of their metabolism. However, the subsequent effects of increasing betaine (a single-carbon donor) are relatively unknown as to their effects on circulating concentrations of metabolites and fatty acids. Changing dietary fatty acids increased circulating concentrations of those dietary fatty acids and this response (the subsequent increase in circulating concentration to a daily fatty acid intake) was increased for EPA with dietary betaine. Dietary betaine also reduced a number of xenobiotics, an effect which shows a positive influence on gut microbiota.

**Abstract:**

In order to evaluate the interaction of betaine and n-3 PUFA in foods consumed by the dog, six extruded dry foods were formulated. The control food had no specific source of added betaine or n-3 fatty acids, while the test foods were supplemented with betaine, flax or fish oil in a 2 × 3 factorial design (no added n-3 source, added flax, added menhaden fish oil, and all with or without added betaine). Forty eight adult dogs were used in this study. All dogs were assigned to one of the six dietary treatments and consumed that food for the length of the 60-day study. Blood was analyzed for metabolomics (plasma), fatty acids and selected health-related analytes (serum) at the beginning and the end of the study. Added dietary betaine increased single-carbon metabolites (betaine, dimethyl glycine, methionine and N-methylalanine), decreased xenobiotics (stachydrine, N-acetyl-S-allyl-L-cysteine, 4-vinylguaiacol sulfate, pyrraline, 3-indoleglyoxylic acid, N-methylpipecolate and ectoine) and enhanced the production of eicosapentaenoic acid (EPA). Dietary betaine also decreased the concentration of circulating carnitine and a number of carnitine-containing moieties. The addition of the n-3 fatty acids alpha-linolenic, EPA and docosahexaenoic acid (DHA) increased their respective circulating concentrations as well as those of many subsequent moieties containing these fatty acids. The addition of alpha-linolenic acid increased the concentration of EPA when expressed as a ratio of EPA consumed.

## 1. Introduction

Polyunsaturated fatty acids (PUFA) are defined by the position of their n-terminal double bond and the number of double bonds. In the n-3 family of PUFA, there is considerable interest in the effects of increasing dietary concentrations and their subsequent influence on whole body immune response and metabolism. The immune modulating ability of increased dietary alpha linolenic acid (αLA), EPA and DHA has recently been reviewed [1]. These have been shown to influence mononuclear cell proliferation [2], enhance the atopic dermatitis scores (αLA, EPA and DHA) [3] and aid in the management of arthritis (EPA and DHA) [4,5,6,7]. We previously reported that dogs had an increased concentration of EPA and DHA after consuming foods with αLA that had below measurable levels of EPA and DHA [8].

The ability of the dog to elongate and desaturate fatty acids within the n-3 family of fatty acids is controversial [9,10] With regard to the n-6 family, it is known that linoleic acid is adequately converted to arachidonic acid so that the dog does not require preformed dietary arachidonic acid [11]. However, the maximum velocity of n-3 vs. n-6 substrates of the Δ-6 desaturase which acts upon αLA is such that it preferentially desaturates n-6 fatty acids. This is why when studying its activity, it was concluded that “*The findings also suggest that high levels of αLA supplementation may be necessary to exceed the Δ-desaturase Km for this substrate and to significantly affect physiological levels of (n-3) long chain PUFA in the dog*” [12]. This need for a high dietary supplementation of αLA to influence the EPA concentration may explain why it was concluded that there was a very limited conversion of αLA to EPA and DHA [13].

Betaine is known to impact the fatty acid metabolism of several species of animals. Dietary supplementation with betaine increases lipolysis in geese to reduce the adipose depot size [14] and counteracts the suppressive effect of insulin on lipolysis in porcine adipose explants [15]. Betaine may reapportion fat stores from adipose to muscle where they can be used as fuel. Supplementation in pigs increases gene expression associated with fatty acid uptake and increases fatty acid levels in muscle, while at the same time increasing the expression of genes involved in fatty acid oxidation [16]. There is selectivity to the action of betaine to increase intramuscular fatty acids, as supplementation in chickens increases the ratio of poly- and monounsaturated to saturated fatty acids [17]. (Conversely, increasing dietary polyunsaturated fatty acids in cats have been shown to decrease the circulating betaine status [18].

Aside from its impact on fatty acid metabolism, betaine serves two basic metabolic roles, functioning both as a methyl donor in single-carbon metabolism and an osmolyte, with reports of it being both an antioxidant and anti-inflammatory in pigs, rats, poultry and humans [19,20]. Betaine’s role as an osmolyte is especially important in kidney health and function. The role of betaine in aiding in the management of kidney disease is to offset the harm caused by hypertonicity—a role which has been extensively studied in cell cultures, where hypertonicity-induced cell death was reduced by betaine in Madin–Darby canine kidney cells [21]. As a donor of a methyl group in single-carbon metabolism, betaine donates the carbon which methylates homocysteine to methionine producing dimethyl glycine through the activity of the enzyme betaine–homocysteine methyltransferase. This is essential as methionine is the donor for many methylation reactions through S-adenosyl methionine (SAM). In dogs with naturally occurring renal disease, a food with enhanced dietary betaine (and increased soluble fiber) has been shown to decrease creatinine (increased creatinine is a sign of renal insufficiency), while increasing the concentration of circulating methionine, suggesting that in dogs with renal disease, it has both roles of a protective osmolyte and a methyl donor [22]. The homocysteine to methionine transformation and accompanying increase in SAM was reported to be one of the active methods for the antioxidant effect of betaine, which was shown to not be dependent on specific antioxidant enzymes [23]. The role of betaine in moderating the immune response has recently been reviewed [24]. In this review it was concluded that much of the immune modulation was through NF-κB and its downstream regulated cytokines.

This study uses dietary fish oil, flax and betaine in order to study the effect of changing the consumption of PUFA, betaine and their interaction on circulating fatty acids, lipids, one-carbon metabolites and xenobiotics.

## 2. Materials and Methods

The Institutional Animal Care and Use Committee, Hill’s Pet Nutrition, Inc., Topeka, KS, USA, approved this protocol (permit # CP832.0.0.0-A-F-MULTI-ADH -MUL-407-OTH), which met the standards of the guides for the care and use of laboratory animals from the National Institutes of Health, US National Research Council and US Public Health Service. All dogs were cared for by animal care research technicians who were masked to the group identity during sample collection. Additionally, all sample analyses and animal care were completed by the research team who did not know which treatment was assigned to the animal or the sample. All dogs were owned by the commercial funders. Forty eight dogs (forty beagles and eight mixed breed) were used in this study, and all dogs completed the study. During the 14-day washout period, all dogs were fed the same pretrial food, which was complete and balanced by the Association of American Feed Control Officials (AAFCO) regulations for canine adult maintenance. Dogs were then assigned by age, weight, gender and breed to either control food or one of five treatment foods and consumed that food exclusively for 60 days. Foods were formulated with and without added betaine and PUFA sources: flax (to increase dietary 18 carbon n-3 fatty acids, specifically α-linolenic acid—αLA) and fish oil (to increase 20+ carbon n-3 fatty acids). The control food had no specific source of betaine or long-chain fatty acids. As feed, the control food analyzed 451 betaine (mg/kg), 0.20% αLA and below quantifiable limits for EPA and DHA (<0.02%). Test foods were formulated as follows: control food with added flax for αLA (0.79%), control with added fish oil as a source of EPA and DHA (0.165 and 0.110%, respectively), control with added betaine (5820 mg/kg), control with both added betaine and αLA (4660 mg/kg and 0.83%, respectively), and control with added betaine, EPA and DHA (5570 mg/kg and 0.155%, 0.105%, respectively). Phlebotomy was completed after a 23 h fast. Global metabolomics were measured at the beginning and end of the 60-day test period. Clinical blood chemistry was also completed on serum at the same time points. Clinical blood analysis was completed on A COBAS c501 module (Roche Diagnostics Corporation, Indianapolis, IN, USA). The metabolomics analysis of plasma was performed by Metabolon (Morrisville, NC, USA) as previously described [18,25]. Fatty acid nutrient analysis of the food was completed by a commercial laboratory (Eurofins Scientific, Inc., Des Moines, IA, USA). The circulating fatty acid (FA) composition in the plasma was determined through separating the FA esters with gas chromatography as previously described [26]. A ratio was calculated for the relationship between individual circulating fatty acids and the daily dietary intake of that respective fatty acid, and was used to assess the effect of dietary betaine and PUFA source on this parameter. This ratio was calculated by dividing the change in circulating concentration of the specific fatty acid during the study (final–initial) by the daily intake of that fatty acid.

All dogs were healthy as shown by clinical indices and physical evaluations. Dogs were housed in rooms with natural lighting access, and given opportunity for enrichment through access to other dogs, care personnel, toys, as well as outside grassed and covered exercise areas. Food intake of a controlled amount was offered daily in an amount calculated to maintain body weight.

All foods were complete and balanced for adult maintenance according to AAFCO. Foods were extruded and contained 11.7% chicken meal, 11.7% grain sorghum, 11.7% red wheat, 10.1% barley, 7.3% oat groats, 5% beat pulp, 4.5% palatability enhancer, 3% corn gluten meal and 2% soybean oil. All foods contained 24.2% whole corn except the flax supplemented foods, where the 3% flax addition was substituted for corn. All foods contained 5.7% pork fat except the foods supplemented with fish oil, where the 1% fish oil was substituted for pork fat. All foods had 3.1% added vitamins and minerals supplying 300 mg/kg L-carnitine and 400 mg/kg taurine.

In order to evaluate the result of treatment on study response criteria (serum biochemical profiles, body weight, and fatty acid composition) the statistical analysis was completed using SAS 9.4 Proc GLM (SAS Institute, Cary, NC, USA). Initial and final values were used in a 2 (with or without added betaine) by 3 (no added PUFA, increased PUFA with flax, increased PUFA with fish oil) factorial analysis. Betaine and PUFA source and their interaction were used as independent variables. An α value for *p* of ≤0.05 was used to reject the null hypothesis that there was no difference between treatments at each time point for the non-metabolomic data. For the analysis of the metabolomics data, the change from the initial value of each pet was used as the response criteria. For the metabolomics data, the change over time or the difference between treatment groups at the end of the study were concluded to be significant if there was a *p* value <0.05 and a false discovery rate correction q value <0.1. In the metabolomics data analysis, log-transformation was completed before analysis and back transformed for inclusion in analysis.

## 3. Results

The analytical food results are shown in Table 1 and differed in the amount and source of PUFA and betaine. Without added betaine, the foods averaged 472 mg/kg, and with added betaine 5350 mg/kg. Without added flax, the foods averaged 0.23% αLA, and with flax they averaged 0.81%. Without added fish oil, the foods had EPA and DHA concentrations below <0.02, and with added fish oil they averaged 0.16% and 0.11% (EPA and DHA, respectively).

There was no difference in body weight at the beginning of the study between treatments and no difference in body weight at the end of the study between treatments. Additionally, body weight did not change within each group during the course of the trial. Treatment also did not have a significant effect on food intake or in final concentrations of circulating albumin, total protein, urea, creatinine, triglycerides or cholesterol (*p* > 0.05, Table 2). There was a change in concentration (*p* < 0.05) over the feeding period as influenced by treatment in only one of the response variables shown in Table 2. This change was in cholesterol, which, although still within the colony normal for adult dogs, was increased in the dogs fed foods containing betaine as compared to those consuming foods unsupplemented with betaine. This was the main effect of betaine (*p* < 0.01).

Dietary betaine and PUFA source resulted in a number of changes (*p* < 0.05) in circulating fatty acids (Table 3). Increasing αLA through the addition of dietary flax resulted in an increase in αLA and the response variable calculated by change in the circulating αLA concentration divided by the daily αLA intake. The dietary αLA intake also resulted in an increase in the circulating concentration of total n-3 fatty acids and an intermediate concentration of EPA and DPA (between no extra added PUFA and increased PUFA with fish oil). There was a specific effect of dietary αLA on the ratio of EPA (circulating EPA change divided by the daily intake) with dietary αLA exceeding both the PUFA unsupplemented control foods and the foods with added fish oil. The fish oil addition increased the circulating concentration of EPA, DPA and DHA, as well as the ratio of EPA when compared to the unsupplemented control food. The addition of dietary betaine increased the circulating concentration of the n-3 fatty acid total, DPA and the ratio of circulating EPA to EPA intake. There was no effect (*p* > 0.05) of dietary betaine or polyunsaturated source on the circulating concentration of the saturated and monounsaturated fatty acids (Table 4).

The metabolomics analysis was nontargeted and detected 749 metabolites. Only those analytes which changed over time (*p* < 0.05, q < 0.1) in at least one treatment and were also significantly changed between treatments at the end of the study (*p* < 0.05, q < 0.1) were included in Table 5. There were 81 metabolites identified by this method. The addition of betaine resulted in increased concentrations of a number of single-carbon metabolites (betaine, dimethyl glycine, methionine and N-methylalanine) as compared to the concentration before consuming betaine. Dietary betaine resulted in the most number of changed metabolites containing carnitine moieties, which were most frequently reduced during the feeding period. Betaine inclusion appeared to be responsible for the reduction in all significantly changed xenobiotic compounds reduced in the dogs while eating the test foods. The addition of the specific dietary PUFA resulted in an increased concentration of that fatty acid while consuming that food or the lipids which contained the enhanced dietary fatty acid. The consumption of dietary fish oil (and the associated increased n-3 fatty acid consumption) led to reduced concentrations of arachidonic acid lipids.

## 4. Discussion

It has been well documented that feeding increased levels of dietary EPA and DHA results in increased circulating concentrations of these fatty acids [11,27]. This increased concentration of circulating fatty acids was associated with changes in the cell membrane fatty acid composition and subsequent reduction in inflammatory prostaglandins, thromboxanes and leukotrienes in dogs [28,29]. This study showed that the efficiency of the conversion of dietary EPA (the ratio of change in the circulating concentration to intake) was increased by dietary betaine. Although this study did not evaluate the mechanism which resulted in this increase, it was, hypothetically, the result of a change in fatty acid turnover as suggested by an effect on lipolysis [14,15]; or by a metabolism associated with fatty acid turnover [16,17]. This (dietary betaine increasing circulating EPA) was also recently shown in quail [30]. This supports dietary betaine as a novel way of increasing circulating EPA in the dog and maximizing the benefit of increased dietary EPA. These data also showed that increasing dietary αLA not only resulted in an increased circulating concentration of αLA, but also resulted in an increased ratio of circulaiting EPA to dietary EPA. Although it was not a tested mechanism, the data could be explained by a more efficient transfer of food nutrient to circulation through an increased absorption or reduced loss through the fatty acid metabolism. However, the more likely explanation is that of the elongation and desaturation of αLA to EPA. In humans, it has been estimated that there is an 8% conversion of αLA to EPA [31]. It seems likely that the conversion of dietary αLA to circulating EPA in the dog is also reduced (as compared to the conversion of dietary EPA to circulating EPA) in a similar fashion as seen in humans, as the higher dietary intake of αLA (as compared to EPA) still resulted in a final circulating EPA concentration of 27% of the EPA-supplemented dogs. A relationship has been detected between αLA and immune modulation in rats [32], humans [33,34] and swine [35]. Although it was the result of the direct effect of this dietary incorporation PUFA, it was likely to be enhanced by the partial conversion of αLA to EPA, thereby having some of its immune modulating effects through the elongation and desaturation to EPA.

The dietary addition of the individual PUFA resulted in an increased concentration of a number of metabolites of that specific fatty acid (e.g., linolenoylcarnitine, 1-palmitoyl-2-docosahexaenoyl-GPC, 1-palmitoyl-2-docosahexaenoyl-GPE and 1-stearoyl-2-docosahexaenoyl-GPE), which indicated the degree to which the fatty acid interventions in the current study manifested in the expected flow of fatty acids into complex lipids. This was in contrast to the observation that dogs consuming the increased EPA and DHA had reduced arachidonic acid compounds (e.g., 1-stearoyl-2-arachidonoyl-GPC and 1-oleoyl-2-arachidonoyl-GPE). This decrease, although not seen in circulating arachidonic acid, suggested there was a competitive inhibition of arachidonic acid incorporation into complex membrane lipids by EPA and/or DHA. Betaine-supplemented foods had a reduction in a number of lipid metabolites, especially carnitine fatty acid moieties. This may be the result of a shift between circulating and organ-bound carnitine as it was previously shown that dietary betaine reduced the severity of the fatty liver response to high-fat food by reversing the inhibition of the carnitine palmitoyltransferase gene expression in the liver that had resulted from a high-fat diet consumption in rats [36]. This conclusion is supported by the report that dietary betaine increased acylcarnitines in mice liver and muscle [37]. However, these data cannot rule out a reduction in total carnitine (as seen in the reduction in circulating carnitine) and the subsequent reduction in carnitine-bound fatty acids. Although more research is needed, a review of the effect of betaine on body composition concluded that the “*increase in intramuscular carnitine would suggest that betaine may increase carnitine palmitoyltransferase I-mediated free fatty acid translocation into the mitochondria and β-oxidation*” [38].

These data show that the dog, as expected [20], uses betaine as a methyl donor, which influences dimethyl glycine, and methionine. The increase in circulating cholesterol has also previously been reported in humans in a recent review, which reported an enhancement in single-carbon metabolites as well as an increase in circulating cholesterol [39]. This shift in single-carbon metabolism has the likelihood of increasing the antioxidant capacity of the dog, similar to what has previously been reported in the chicken [40,41], grass carp [42], domestic cat [43], cattle [44] and rabbit [45]. The increased cholesterol concentration, although similar to that seen in humans in response to betaine intake, did not change concentration above that which has been reported for normal healthy dogs [46].

The inclusion of dietary betaine reduced the circulating concentration of a number of microbiota-related metabolites and increased circulating methionine. We previously saw that circulating betaine and microbial-derived metabolites were both influenced by fatty acids in cats [18]. However, there was no clear relationship between the directionality of these changes and the animals’ fatty acid intake. In this study, of the microbial xenobiotics that were influenced, there was a significant overall reduction in those xenobiotics in the dogs consuming enhanced dietary betaine. This reduction in the specific xenobiotics suggests a positive health effect. For example, the advanced glycation end product pyrraline has been repeatedly shown to be linked to, and may even be responsible for, changes associated with aging that may negatively affect health and longevity. This may be performed through direct action or through secondary intermediates that are toxic, causing oxidative damage and reacting with proteins to reduce their function [47]. It is possible that the reduction in the observed xenobiotics was associated with a reduction in renal toxic microbial metabolites. These changes in the xenobiotics observed with the addition of betaine support the understanding of the positive effect seen in feeding dietary betaine to dogs with early renal disease [22].

## 5. Conclusions

There was an interaction between single-carbon and fat metabolism as influenced by dietary betaine and PUFA metabolism. This influence and interaction may be a benefit to the dog in enhancing the positive immune modulating effects of n-3 fatty acids. The xenobiotic results of dietary betaine and the interactive influence of betaine and PUFA suggest a possible health benefit, which merits further investigations.

## Figures and Tables

**Table 1 animals-12-00768-t001:** Food composition of pretrial and test foods (g/100 g of food as fed unless otherwise stated).

Analyte	Control (and Pretrial)	Control + Flax	Control + Fish Oil	Control + Betaine	Control + Flax + Betaine	Control + Fish Oil + Betaine
Moisture	8.45	8.98	8.21	8.52	8.61	8.67
Protein	17.63	20.19	17.85	18.68	18.00	18.94
Fat	13.26	15.18	13.97	14.38	14.61	13.50
Atwater energy ^€^ (kcal/kg)	3633	3698	3670	3675	3693	3622
Ash	4.48	4.59	4.58	4.77	4.56	4.93
Crude fiber	2.20	2.45	2.30	2.25	2.20	2.20
Calcium	0.74	0.78	0.76	0.79	0.70	0.82
Phosphorus	0.62	0.66	0.65	0.68	0.64	0.67
Sodium	0.24	0.24	0.25	0.27	0.24	0.27
Betaine (mg/kg)	451	435	529	5820	4660	5570
Capric acid (10:0)	<0.02	<0.02	<0.02	<0.02	<0.02	<0.02
Lauric acid (12:0)	<0.02	<0.02	<0.02	<0.02	<0.02	<0.02
Myristic acid (14:0)	0.09	0.11	0.16	0.11	0.10	0.15
Palmitic acid (16:0)	2.31	2.68	2.53	2.77	2.55	2.45
Palmitoleic acid (16:1)	0.25	0.28	0.34	0.30	0.27	0.33
Stearic acid (18:0)	1.02	1.17	1.01	1.18	1.10	0.98
Oleic acid (18:1)	3.79	4.44	3.87	4.45	4.25	3.74
Arachidic acid (20:0)	0.03	0.03	0.03	0.03	0.03	0.03
Linoleic acid (18:2 (n-6))	2.75	3.34	3.02	3.34	3.24	2.88
αLinolenic acid (18:3 (n-3))	0.20	0.79	0.25	0.25	0.83	0.22
Arachidonic acid (20:4 (n-6))	0.02	0.035	0.045	0.04	0.04	0.045
EPA (20:5 (n-3))	<0.02	<0.02	0.165	<0.02	<0.02	0.155
DPA (22:5 (n-3))	<0.02	<0.02	0.02	<0.02	<0.02	0.02
DHA (22:6 (n-3))	<0.02	<0.02	0.110	<0.02	<0.02	0.105
SFA ^£^	3.53	4.07	3.80	4.19	3.76	3.70
MUFA ^¥^	4.17	4.88	4.37	4.92	4.66	4.24
PUFA ^π^	3.04	4.07	3.73	3.74	4.20	3.53
(n-6) FA ^Ω^	2.83	3.25	3.15	3.46	3.35	3.00
(n-3) FA ^θ^	0.21	0.82	0.58	0.28	0.85	0.53
(n-6):(n-3) ratio	13.5	4.0	5.4	12.4	3.9	5.7

^€^ Calculated by using 3.5, 8.5 and 3.5 kcal/g for protein, fat and 3.5 nitrogen-free extract. ^£^ Saturated fatty acids, the sum calculated: 8:0 + 10:0 + 11:0 + 12:0 + 14:0 + 15:0 + 16:0 + 17:0 + 18:0 + 20:0 + 22:0 + 24:0. ^¥^ Monounsaturated fatty acids, the sum calculated: 14:1 + 15:1 + 16:1 + 17:1 + 18:1 + 20:1 + 22:1 + 24:1. ^π^ Polyunsaturated fatty acids, the sum calculated: 18:2(n-6) + 18:3(n-6) + 18:3(n-3) + 18:4(n-3) + 20:2(n-6) + 20:3(n-6) + 20:3(n-3) + 20:4(n-6) + 20:4(n-3) + 20:5(n-3) + 21:5(n-3) + 22:2(n-6) + 22:4(n-6) + 22:5(n-6) + 22:5(n-3) + 22:6(n-3). ^Ω^ Sum of the (n-6) fatty acids. ^θ^ Sum of the (n-3) fatty acids.

**Table 2 animals-12-00768-t002:** The influence of dietary betaine and fatty acids on body weight and selected serum biochemistries (values are ls means ± standard errors).

Analyte	Control	Control + Flax	Control + Fish Oil	Control + Betaine	Control + Flax + Betaine	Control + Fish Oil + Betaine
Body Weight (kg) Initial	11.4 ± 0.91	11.9 ± 0.91	10.9 ± 0.91	11.5 ± 1.05	11.1 ± 0.91	11.0 ± 0.91
Body Weight (kg) Final	11.7 ± 0.91	11.5 ± 0.91	10.8 ± 0.91	11.2 ± 1.05	11.4 ± 0.91	11.0 ± 0.91
Food intake (g/day)	229 ± 16.2	196 ± 16.2	204 ± 16.2	210 ± 16.2	226 ± 16.2	200 ± 16.2
Albumin (mg/dL) Initial	3.59 ± 0.08	3.56 ± 0.08	3.36 ± 0.08	3.55 ± 0.08	3.50 ± 0.08	3.64 ± 0.08
Albumin (mg/dL) Final	3.55 ± 0.09	3.57 ± 0.09	3.38 ± 0.09	3.41 ± 0.09	3.42 ± 0.09	3.64 ± 0.09
Total Protein (mg/dL) Initial	5.62 ± 0.10	5.68 ± 0.10	5.29 ± 0.10	5.49 ± 0.10	5.54 ± 0.10	5.61 ± 0.10
Total Protein (mg/dL) Final	5.61 ± 0.12	5.78 ± 0.12	5.46 ± 0.12	5.51 ± 0.12	5.50 ± 0.12	5.79 ± 0.12
Urea Nitrogen (mg/dL) Initial	10.4 ± 0.8	10.9 ± 0.8	10.0 ± 0.8	10.3 ± 0.8	10.3 ± 0.8	11.7 ± 0.8
Urea Nitrogen (mg/dL) Final	11.3 ± 0.7	9.9 ± 0.7	10.3 ± 0.7	9.8 ± 0.7	9.8 ± 0.7	11.4 ± 0.7
Creatinine (mg/dL) Initial	0.73 ± 0.04	0.75 ± 0.04	0.67 ± 0.04	0.81 ± 0.04	0.71 ± 0.04	0.72 ± 0.04
Creatinine (mg/dL) Final	0.78 ± 0.04	0.68 ± 0.04	0.69 ± 0.04	0.81 ± 0.04	0.68 ± 0.04	0.73 ± 0.04
Triglycerides (mg/dL) Initial	57.0 ± 17.3	75.5 ± 17.3	62.4 ± 14.7	69.9 ± 14.7	67.1 ± 14.7	80.2 ± 14.7
Triglycerides (mg/dL) Final	79.3 ± 14.7	69.8 ± 14.7	62.4 ± 14.7	60.4 ± 17.3	55.5 ± 17.3	92.4 ± 17.3
Cholesterol (mg/dL) Initial	199.1 ± 16.9	204.5 ± 16.9	203.5 ± 16.9	209.8 ± 16.9	205.0 ± 16.9	200.4 ± 16.9
Cholesterol (mg/dL) ^¥^ Final	196.4 ± 17.9	209.1 ± 17.9	191.8 ± 17.9	220.0 ± 17.9	220.9 ± 17.9	225.8 ± 17.9

^¥^ Although initial and final values did not differ by treatment, there was a main effect of dietary betaine which resulted in a significantly higher delta (final–initial) concentration.

**Table 3 animals-12-00768-t003:** The influence of dietary betaine and fatty acids on circulating concentration (mg/dL) of polyunsaturated fatty acids or ratio of change in circulating concentration over the course of the study (final–initial) divided by daily intake (µg/dL/mg daily intake). Values are ls means ± standard errors.

Analyte	Control	Control + Flax	Control + Fish Oil	Control + Betaine	Control + Flax + Betaine	Control + Fish Oil + Betaine	F-Test *p* Value ^Ω^
LA (18:2 (n-6)) Initial	51.6 ± 3.1	48.4 ± 3.1	49.6 ± 3.1	52.3 ± 3.1	49.6 ± 3.1	49.7 ± 3.1	0.95
LA (18:2 (n-6)) Final	52.4 ± 3.2	52.4 ± 3.2	50.5 ± 3.2	55.1 ± 3.2	54.1 ± 3.2	52.5 ± 3.2	0.94
αLA (18:3 (n-3)) Initial	1.4 ± O.1	1.5 ± O.1	1.2 ± O.1	1.4 ± O.1	1.3 ± O.1	1.4 ± O.1	0.30
αLA (18:3 (n-3)) Final	1.2 ± O.1 ^a^	2.4 ± O.1 ^b^	1.1 ± O.1 ^a^	1.3 ± O.1 ^a^	2.5 ± O.1 ^b^	1.5 ± O.1 ^a^	<0.01 P
αLA ratio	−0.4 ± 0.2 ^a^	0.6 ± 0.2 ^b^	−0.1 ± 0.2 ^a^	−0.1 ± 0.2 ^a^	0.7 ± 0.2 ^b^	0.1 ± 0.2 ^a^	0.01 P
ARA (20:4 (n-6)) Initial	54.7 ± 3.6	51.4 ± 3.6	50.8 ± 3.6	50.6 ± 3.6	51.1 ± 3.6	53.6 ± 3.6	0.95
ARA (20:4 (n-6)) Final	57.0 ± 3.7	53.5 ± 3.7	47.8 ± 3.7	57.0 ± 3.7	55.2 ± 3.7	50.8 ± 3.7	0.46
EPA (20:5 (n-3)) Initial	0.9 ± 0.1	0.9 ± 0.1	0.9 ± 0.1	0.9 ± 0.1	0.9 ± 0.1	0.9 ± 0.1	0.99
EPA (20:5 (n-3)) Final	0.5 ± 0.4 ^a^	1.3 ± 0.4 ^a,b^	5.1 ± 0.4 ^b^	0.7 ± 0.4 ^a^	1.6 ± 0.4 ^a,b^	5.5 ± 0.4 ^b^	<0.01 P
EPA ratio	−30.6 ± 7.1 ^d^	41.9 ± 7.1 ^b^	12.5 ± 7.1 ^c^	−20.2 ± 7.1 ^d^	71.7 ± 7.1 ^a^	15.2 ± 7.1 ^c^	<0.01 P*B
DPA (22:5 (n-3)) Initial	7.6 ± 0.7	7.0 ± 0.7	7.5 ± 0.7	7.4 ± 0.7	7.3 ± 0.7	7.9 ± 0.7	0.95
DPA (22:5 (n-3)) Final	6.1 ± 0.9 ^a,d^	8.9 ± 0.9 ^a,b,d^	10.2 ± 0.9 ^b,d^	7.2 ± 0.9 ^a,e^	10.5 ± 0.9 ^a,b,e^	12.8 ± 0.9 ^b,e^	<0.01 P,B
DHA (22:6 (n-3)) Initial	2.6 ± 0.4	3.6 ± 0.4	2.8 ± 0.4	2.7 ± 0.4	2.9 ± 0.4	3.5 ± 0.4	0.72
DHA (22:6 (n-3)) Final	1.7 ± 0.9 ^a^	2.4 ± 0.9 ^a^	9.5 ± 0.9 ^b^	2.0 ± 0.9 ^a^	2.1 ± 0.9 ^a^	11.5 ± 0.9 ^b^	<0.01 P
DHA ratio	−73.2 ± 20.4 ^c^	−13.0 ± 20.4 ^b^	28.4 ± 20.4 ^a^	−68.3 ± 20.4 ^c^	−67.8 ± 20.4 ^c^	39.1 ± 20.4 ^a^	<0.01 P
Sum of n-3 ^£^ Initial	12.5 ± 0.9	12.4 ± 0.9	13.0 ± 0.9	12.0 ± 0.9	13.4 ± 0.9	13.7 ± 0.9	0.98
Sum of n-3 ^£^ Final	9.6 ± 1.8 ^a,d^	14.9 ± 1.8 ^b,d^	25.7 ± 1.8 ^c,d^	11.1 ± 1.8 ^a,e^	16.7 ± 1.8 ^b,e^	31.2 ± 1.8 ^c,e^	<0.01 P,B
Sum of n-6 ^£^ Initial	113.3 ± 6.5	109.7 ± 6.5	105.9 ± 6.5	108.0 ± 6.5	107.0 ± 6.5	110.0 ± 6.5	0.56
Sum of n-6 ^θ^ Final	117.4 ± 6.7	120.8 ± 6.7	112.6 ± 6.7	116.8 ± 6.7	104.1 ± 6.7	109.9 ± 6.7	0.78
Sum of PUFA ^¥^ Initial	125.8 ± 7.3	122.0 ± 7.3	118.9 ± 7.3	120.4 ± 7.3	119.4 ± 7.3	123.7 ± 7.3	0.98
Sum of PUFA ^¥^ Final	127.0 ± 7.6	131.9 ± 7.6	127.6 ± 7.6	133.5 ± 7.6	129.8 ± 7.6	141.1 ± 7.6	0.79

^a,b,c,d,e^ Means with different superscripts are different (*p* ≤ 0.05). ^£^ Sum of n-3 fatty acids defined above. ^θ^ Sum of n-6 fatty acids defined above. ^¥^ Sum of the polyunsaturated fatty acids: 18:2(n-6) + 18:3(n-6) + 18:3(n-3) + 20:2(n-6) + 20:3(n-6) + 20:3(n-3) + 20:4(n-6) + 20:4(n-3) + 20:5(n-3) + 21:5(n-3) + 22:5(n-3) + 22:6(n-3). ^Ω^ The *p* value is for the f test of the model, P is a main effect of PUFA source, B a main effect of betaine, P*B a significant effect of the interaction of betaine and polyunsaturated fat source.

**Table 4 animals-12-00768-t004:** Circulating concentration (mg/dL) of saturated and monounsaturated fatty acid profiles (values are ls means ± standard errors).

Analyte	Control	Control + Flax	Control + Fish Oil	Control + Betaine	Control + Flax + Betaine	Control + Fish Oil + Betaine	F-Test *p* Value ^Ω^
Myristic acid (14:0) Initial	0.75 ± 0.09	0.71 ± 0.09	0.63 ± 0.09	0.73 ± 0.09	0.60 ± 0.09	0.63 ± 0.09	0.73
Myristic acid (14:0) Final	0.69 ± 0.07	0.54 ± 0.07	0.59 ± 0.07	0.61 ± 0.07	0.47 ± 0.07	0.57 ± 0.07	0.33
Palmitic acid (16:0) Initial	37.7 ± 2.2	33.9 ± 2.2	35.8 ± 2.2	36.4 ± 2.2	36.4 ± 2.2	36.2 ± 2.2	0.89
Palmitic acid (16:0) Final	39.2 ± 2.2	37.3 ± 2.2	37.8 ± 2.2	40.0 ± 2.2	36.6 ± 2.2	39.3 ± 2.2	0.80
Palmitoleic acid (16:1) Initial	1.9 ± 0.16	2.4 ± 0.16	1.5 ± 0.16	1.6 ± 0.16	1.6 ± 0.16	2.1 ± 0.16	<0.01 P*B
Palmitoleic acid (16:1) Final	2.1 ± 0.19	2.0 ± 0.19	1.5 ± 0.19	1.8 ± 0.19	1.5 ± 0.19	1.9 ± 0.19	0.20
Stearic acid (18:0) Initial	63.2 ± 3.9	59.0 ± 3.9	61.3 ± 3.9	59.1 ± 3.9	60.1 ± 3.9	62.3 ± 3.9	0.96
Stearic acid (18:0) Final	65.4 ± 4.3	68.2 ± 4.3	68.1 ± 4.3	68.5 ± 4.3	79.1 ± 4.3	76.1 ± 4.3	0.70
Oleic acid (18:1) Initial	22.8 ± 1.1	25.2 ± 1.1	20.1 ± 1.1	22.3 ± 1.1	23.3 ± 1.1	23.5 ± 1.1	0.09
Oleic acid (18:1) Final	25.7 ± 1.5	25.3 ± 1.5	20.5 ± 1.5	24.3 ± 1.5	23.8 ± 1.5	23.3 ± 1.5	0.19

^Ω^ The *p* value is for the f test of the model, P*B appears as there was a significant effect of assignment to foods with betaine and polyunsaturated fat source.

**Table 5 animals-12-00768-t005:** The influence of dietary betaine and fatty acids on the ratio (initial concentration/final concentration) of selected analytes from metabolomics dogs eating control food or foods supplemented with flax, fish oil and/or betaine.

Biochemical	Control	Control + Flax	Control + Fish Oil	Control + Betaine	Control + Flax + Betaine	Control + Fish Oil + Betaine
**Amino acid metabolites**						
Sarcosine	0.77	0.75	0.78	1.01	0.98	1.07
Dimethylglycine	1.01	1	1.01	1.44	1.66	1.73
Betaine	0.9	0.9	0.93	1.84	1.74	1.8
N-methylalanine	0.78	0.74	0.53	1.45	1.67	1.62
N,N-dimethylalanine	1.27	0.96	1.33	0.13	0.19	0.1
pyroglutamine	0.68	0.82	0.61	0.43	0.57	0.46
tryptophan betaine	1.04	0.69	0.83	0.43	0.35	0.41
isovalerylcarnitine (C5)	1.08	0.87	1.06	0.48	0.62	0.61
2-methylbutyrylcarnitine (C5)	1.03	0.83	1.04	0.6	0.59	0.67
isobutyrylcarnitine (C4)	1.12	0.91	1.24	0.62	0.6	0.65
methionine	1.02	1	1.01	1.08	1.13	1.2
alpha-ketobutyrate	1.14	1.57	1.3	0.92	1.18	1.25
cysteine sulfinic acid	0.8	1.02	0.86	0.87	1.53	0.91
taurine	1.06	0.99	0.92	1.21	1.25	0.75
N-methylproline	1.12	0.77	0.96	0.62	0.72	0.58
**Lipids**						
stearidonate (18:4 n3)	0.71	0.62	2.52	0.67	0.85	2.03
eicosapentaenoate (EPA; 20:5 n3)	0.64	0.61	2.87	0.57	0.98	3.23
heneicosapentaenoate (21:5 n3)	0.34	0.39	4.28	0.39	0.43	3.73
docosapentaenoate (n3 DPA; 22:5 n3)	0.93	0.78	1.09	0.72	1.22	1.51
docosahexaenoate (DHA; 22:6 n3)	0.58	0.5	3.49	0.51	0.57	2.99
linolenate (alpha or gamma; (18:3 n3 or 6))	0.85	1.13	0.77	0.8	1.63	0.83
(15 or 16)-methylmargarate (a18:0 or i18:0)	0.86	0.86	0.94	0.59	0.95	1.04
3-hydroxydodecanedioate	0.76	0.87	0.78	0.38	0.54	0.41
butyrylcarnitine (C4)	1.11	0.79	0.95	0.52	0.62	0.66
propionylcarnitine (C3)	1.27	0.85	1.08	0.68	0.77	0.63
acetylcarnitine (C2)	1.18	1.05	1.16	0.7	0.69	0.75
hexanoylcarnitine (C6)	1.13	0.91	1.09	0.64	0.73	0.68
octanoylcarnitine (C8)	0.97	1.02	1.07	0.66	0.72	0.68
decanoylcarnitine (C10)	0.93	1.12	1.27	0.82	0.73	0.61
laurylcarnitine (C12)	0.9	0.97	0.81	0.48	0.54	0.6
myristoylcarnitine (C14)	0.94	0.94	0.8	0.56	0.56	0.71
palmitoylcarnitine (C16)	0.69	1.03	0.8	0.66	0.71	0.63
margaroylcarnitine (C17)	1.04	1.44	0.88	0.66	0.85	0.84
stearoylcarnitine (C18)	1.04	1.17	0.83	0.63	0.97	0.7
arachidoylcarnitine (C20)	1.01	1.07	0.89	0.75	0.92	0.78
cis-4-decenoylcarnitine (C10:1)	1.26	1.17	1.13	0.63	0.6	0.75
5-dodecenoylcarnitine (C12:1)	1.02	1.09	1.08	0.66	0.71	0.72
myristoleoylcarnitine (C14:1)	0.85	1.17	1.01	0.56	0.62	0.63
palmitoleoylcarnitine (C16:1)	1.06	1.07	1.36	0.71	0.74	1.02
eicosenoylcarnitine (C20:1)	1.07	1.01	0.81	0.81	0.75	0.67
linoleoylcarnitine (C18:2)	0.97	1.16	1.04	0.73	0.85	0.81
linolenoylcarnitine (C18:3)	0.71	1.46	0.69	0.6	1.22	0.68
adipoylcarnitine (C6-DC)	1.07	0.77	0.81	0.54	0.55	0.52
(S)-3-hydroxybutyrylcarnitine	1.26	0.9	1.15	0.57	0.61	0.56
carnitine	1.08	0.99	1.06	0.75	0.71	0.75
1-myristoyl-2-palmitoyl-GPC (14:0/16:0)	0.72	0.77	0.75	0.61	0.64	0.79
1-palmitoyl-2-docosahexaenoyl-GPC (16:0/22:6)	0.74	0.68	2.93	0.78	0.71	2.66
1-palmitoleoyl-2-linoleoyl-GPC (16:1/18:2)	0.94	1.08	0.83	0.98	1.1	0.83
1-palmitoleoyl-2-linolenoyl-GPC (16:1/18:3)	0.76	1.2	0.98	0.86	1.46	0.99
1-stearoyl-2-arachidonoyl-GPC (18:0/20:4)	1.03	0.97	0.85	1.03	1.04	0.88
1-stearoyl-2-docosahexaenoyl-GPC (18:0/22:6)	0.68	0.71	3.05	0.78	0.81	2.86
1-oleoyl-2-docosahexaenoyl-GPC (18:1/22:6)	0.8	0.63	2.61	0.75	0.68	2.62
1,2-dilinoleoyl-GPC (18:2/18:2)	0.9	1.16	0.83	0.95	1.17	0.87
1-linoleoyl-2-arachidonoyl-GPC (18:2/20:4 n6) *	0.9	1.02	1	0.95	1.1	1.09
1,2-dilinolenoyl-GPC (18:3/18:3)	0.49	1.88	5.7	0.75	3.01	5.29
1-palmitoyl-2-docosahexaenoyl-GPE (16:0/22:6)	0.74	0.77	3.47	0.86	0.71	3.24
1-stearoyl-2-docosahexaenoyl-GPE (18:0/22:6)	0.51	0.68	5.2	0.71	0.59	4.18
1-oleoyl-2-arachidonoyl-GPE (18:1/20:4)	1.03	1.33	0.68	1.16	1.22	0.7
1-linolenoyl-GPC (18:3)	0.65	1.75	0.82	0.82	1.97	0.86
1-oleoyl-GPE (18:1)	0.91	1.05	0.64	0.89	0.88	0.72
1-linoleoyl-GPE (18:2)	0.96	1.16	0.68	0.95	0.96	0.66
1-(1-enyl-stearoyl)-2-oleoyl-GPE (P-18:0/18:1)	0.89	1.05	0.89	1.02	0.88	0.74
1-(1-enyl-palmitoyl)-2-arachidonoyl-GPC (P-16:0/20:4)	1.08	0.94	0.74	1.1	0.98	0.76
1-(1-enyl-palmitoyl)-2-linoleoyl-GPC (P-16:0/18:2)	1.1	1.08	0.81	1.14	1.1	0.8
sphingomyelin (d18:2/18:1)	0.97	0.95	0.86	1.1	1	0.89
sphingomyelin (d18:2/23:1)	0.95	0.96	1.26	0.96	1.1	1.44
sphingomyelin (d18:1/18:1, d18:2/18:0)	1.11	1.04	0.92	1.18	1	0.93
sphingomyelin (d18:1/20:2, d18:2/20:1, d16:1/22:2)	1.09	0.91	0.71	1.25	0.88	0.87
sphingomyelin (d18:2/21:0, d16:2/23:0)	0.95	1.03	1.15	0.98	1.11	1.38
palmitoyl-sphingosine-phosphoethanolamine (d18:1/16:0)	1.03	1.08	0.67	1.24	1.1	0.68
**Nucleotides, vitamins and cofactors**						
5-methylcytidine	1.04	0.88	0.98	0.94	0.94	0.83
5-hydroxymethyl-2′-deoxycytidine	1.09	0.93	0.96	0.92	0.96	0.84
trigonelline (N’-methylnicotinate)	0.97	0.84	1.18	0.53	0.59	0.53
pantothenate	1.25	1.22	1.2	1.09	1.15	1.09
**Xenobiotics**						
stachydrine	0.61	0.51	0.99	0.56	0.61	0.52
N-acetyl-S-allyl-L-cysteine	1.24	1.05	1.08	0.49	0.45	0.45
4-vinylguaiacol sulfate	0.82	0.84	1.31	0.69	0.4	0.65
pyrraline	1.23	0.58	1.12	0.86	0.74	0.89
3-indoleglyoxylic acid	1.08	0.83	1.03	0.9	1.04	0.84
N-methylpipecolate	0.96	0.95	1.29	0.71	0.63	0.76
ectoine	1.22	1.08	1.13	0.49	0.49	0.49

* This table is the selected group of analytes where there was a *p* ≤ 0.05 and a q ≤ 0.1 for both change during the test feeding period and a significant difference between treatments at the end of the test feeding period. Values were change during the study while eating the listed food. A decline appears as green, while values in red show a significant increase in the ratio (initial concentration/final concentration) of that compound.

## Data Availability

Data is contained in the article or available from the corresponding author.

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
