# Peer review of "Dietary Betaine and Fatty Acids Change Circulating Single-Carbon Metabolites and Fatty Acids in the Dog"

_animals, 2022, doi:10.3390/ani12060768_

Round 1

Reviewer 1 Report

Line 303 - Don't the authors mean "dog" rather than "cat"?

Author Response

Thank you for your review.  You are exactly correct – we meant dog and have changed the text accordingly.

Reviewer 2 Report

General comments

Citations should follow journal guidelines.

I would encourage the authors to upload the full metabolomics data from the different treatments as a supplemental material. For the paper is would be fine it is reported as a ratio. However, for the readers it would be beneficial to have the actual concentration of these different substances as a reference for future studies.

The discussion is short and could be expanded to provide some explanations for the data you presented in this work. What are the implications of the changes you reported for the Nucleotides, vitamins, and co-factors? There is only a discussion for the pyrraline, what about the other xenobiotics you presented on the table? What their changes means to the dogs? Pyrraline was only affected by Flax +Betaine treatment, why do you think that happened?

Specific comments

L16, 26, 82, 291, 292, 293, Table 5 – What is the difference between “xenobiotics” and “xenobioticss”? In some instances, you use one and in others you use the other.

L125-130 – please provide the full ingredient composition of all experimental foods.

L146-149 – seems odd to me that you moved “mg/kg” in the parenthesis and the “%” you didn’t. Also, the unit is not “mg/kg betaine”, but “mg/kg food”

Table 1 – are the values reported here adjusted for the food dry matter? While the foods had very close moisture levels, it would be relevant to the reader to know. “LA”, “ARA”, “DPA”, SFA”, “MUFA” not described previously.

L199-200 – I think you meant “above” instead of “below”

Table 4 – this table could be combined with table 3. Because there are no treatment effects on saturated FAs and the focus of the study was on the PUFAS, maybe you could only show data for the saturated FAs as a whole rather than individual saturated FAs.

Table 5 – the tile of this table needs to be updated. You should make it clear from the title that the values you report here are the ratio initial/final concentration instead of adding this in the footnote of the table. Mainly because of the length of this table, it would be beneficial to the reader to have this information already in the title.

Author Response

We have corrected the citations.

Unfortunately, these data are all semi-quantitative and the actual concentrations of the compounds are not available.

The discussion was expanded as requested.

“xenobiotics” and “xenobioticss – this was an inadvertent spelling error and we have corrected it xenobiotics being the correct term.

Full ingredient composition is now shown in rewrite of lines 78-31.

Lines 93-94 mg/kg now included without parenthesis as requested.

Table 1 was clarified so that it is clearly as fed.  Abbreviations are more clearly defined in the footnotes.

L 199-200 Thank you for pointing this out.  We have corrected the typo.

Table 3 and 4.  I understand the reviewers desire for simplification.  However, as we have used this break in reporting circulating fatty acid compositions in our work it seems best to continue the classification breaks between the PUFA and the saturated/monounsaturated fatty acids.

Table 5 – thank you for your suggestion.  We have enhanced the title for better clarity.

Reviewer 3 Report

The paper is well-written. However, it is unclear why betaine was chosen as the supplement. Its potential benefits in regard to renal health are discussed, however this was not an outcome of this study. 

There are some key details missing in the materials and methods, and the discussion should be expanded upon in certain areas. Specifically the pathways that betaine acted within to result in these observed increases. Also, the conclusions that you the reader should take from this manuscript and the results should be made clearer.  

Results section is largely missing p-values, and references to support some information presented are missing. 

Tables should have more descriptive titles. 

Simple summary: 

Line 8-10 "... intakes have relatively unknown subsequent effects on circulating concentrations of those fatty acids and their metabolites". This first sentence is quite hard to follow. Perhaps change those to said to make it clearer. 

Lines 10-11: "Single carbon metabolism ... is known to be an essential part of their metabolism". This sentence seems redundant, would recommend re-wording. 

Line 13: " Changing dietary acids increased ...." Changing in terms of source or concentration? 

Line 15: "... was increased for EPA with dietary betaine". Recommend changing this to "further increased" for clarity.

Lines 16-17 "... an effect which shows a positive influence on gut microbiota". In said study, or is this speculated based on previous data on other species? 

Abstract:

Line 19: Are the foods extruded or canned? 

Line 22: All adult dogs?

Line 23: Was whole blood analyzed, or serum or plasma? 

Line 26: "xenobioticss" - typo: extra s on send. 

EPA & DHA should be defined. 

Unclear from the abstract what the take-aways or important of this work are. 

Introduction: 

Lines 39-44: in which species?

Lines 44-46: In serum or plasma? 

Lines 47-48: Reference missing

Line 48-50: Reference missing

Lines 59-60: "....reports of it being both an anti-oxidant and anti-inflammatory" - in which species?

Line 63: "cell culture" - what kind of cells? 

Lines 64-68: References missing. 

Paragraph on betaine (lines 58-79): The connection between betaine and PUFAs is not clear. 

Materials & Methods: 

Lines 93-94: What was the pre-trial diet that the dogs consumed? Was it a commercial diet? If so, should be included as a footnote. Also, was the food complete and balanced according to AAFCO?

Lines 94-95: Was BCS of the dogs assessed and considered? 

Line 96: Were the foods extruded or canned? Were they formulated to meet AAFCO? 

Lines 98-99: "The control food has no specific source of betaine or long chain fatty acids analyzed 451 mg/kg betaine..." - has should be had (past tense). Also 451 mg of what? Betaine? If so reads as mg betaine/kg betaine. 

Lines 105-108: Was fasted blood taken? 

Lines 107: Was clinical blood analysis before on whole blood, serum or plasma?

Lines 109 to 110: Repetition with plasma "The metabolomics analysis of plasma was performed by ... of the plasma as previously described..."

Lines 111: "Fatty acid nutrient analysis' .. analysis of diet or blood? Serum or plasma? 

Lines 136-137: Was a post hoc performed to assess differences between treatments? If so, which post hoc was used. 

Line 141: was log-transformed data back-transformed for presentation in the manuscript? 

Results: 

Lines 144-145: "All foods were complete and balanced..." Already included in section above. Repetition.

Lines 146-147: "Without added betaine the foods averaged .." mg of what? Betaine? If so, again reads as mg betaine/kg betaine

Lines 161-164: p-values? 

Line 166: "There was a change in concentration over the feeding period.." Change in concentration of what? 

Lines 167-168: "Cholesterol was increased in the dogs..." Was cholesterol still within reference range?

Table 2 missing p-values. Mentioned above that cholesterol changed. If so, a post-hoc should be performed and included to show differences between treatments. Could consider also presenting table in a different way to make initial vs. final concentrations in a treatment easier to compare between. 

Lines 177-191: P-values missing. 

Table 3: Was a comparison between the initial and final values within a treatment done? Superscripts included, but not included beneath table which post-hoc was used. 

Line 212-213: Were these increased concentrations statistically significant or numerical?

Line 219-221: P-values missing. Were these reduced concentrations in comparison to the other treatments or to initial value? 

Table 5: Were differences between the treatments assessed? Unclear why there is a * next to only pyroglutamine. 

Discussion:

Line 233: "This increased concentration is ..." increased concentration of what?

Lines 234-235 ".. subsequent reduction in inflammatory prostaglandins, thromboxanes ..." Has this been documented in dogs or other species? 

Lines 235-238: "This study showed that the efficiency..." What was the pathway that resulted in this? Unclear how this was achieved by betaine supplementation. 

Line 242: "This could be the result of a more efficient transfer of food nutrient.." How so?

Lines 286-300: What is the biological significance of this? Unclear what the takeaway is in regards to health in dogs. 

Conclusion

Line 303: ".. may be a benefit to the cat". Should say dog?

Acknowledgements blank. 

Author Response

More information regarding the role of betaine as well as increased references were added for clarity.

Materials and Methods were enhanced for clarity.

Statistical p values were added for clarity. 

Titles were changed to a more descriptive title.

Abstract

L 19 rewritten including extruded for clarity

L 22 rewritten to include adult for clarity

L 44-46 plasma and serum added for clarity       

L 48-50 a reference was added for clarity

L 59-60 species added as requested

Expanded the introduction on betaine and PUFA metabolism

Materials and Methods changes stated below:

L 93-94 pretrial food was clearly stated as complete and balanced for adult maintenance according to AAFCO definition.

BCS was not evaluated.

L 98-99 was reworded for clarity as requested.

Fasted blood was taken as the paper states “Phlebotomy was completed after a 23 hour fast”

L 107 blood use was clarified as requested.

L 109-110 the repetitive “of plasma” was removed.  Thank you for pointing out this typo.

L 111 fatty acid analysis was clarified as suggested.

L141 – that is correct, values were back transformed and this is now included for clarity.

L144-147 changed to add clarity as requested.

L161-164 p values added for clarification.

L166-168 changed for clarity.

Table 2.  The individual means in the post hoc tests were not different but there was a main effect of dietary betaine.  Description was changed to make this point more clear.

Table 3 – the superscript used but not included in the footnote was corrected.  Thank you for pointing out our error.

L212-213 statistically significant changes, wording and p values added for clarification.

L 219 221 change was described more clearly for clarification.

Table 5 – the * were removed as they were included in error.  The paper was rewritten to show that  difference between treatments were assessed in order to allow inclusion in the table.

L 234-235 changed for clarity

L235-238 – discussion was added to give the reader some possibilities to consider as a mode of action.

L 242 – discussion was changed as although this is a factual possibility we don’t know of a mechanism that would support that conclusion.

L 286-300 discussion was added to suggest to the reader a basis for the benefit as requested.

Line 303 – typo corrected.  Thank you for pointing it out for us.